# Ceria Quantum Dot Filler-Modified Polymer Electrolytes for Three-Dimensional-Printed Sodium Solid-State Batteries

**DOI:** 10.3390/polym16121707

**Published:** 2024-06-14

**Authors:** Yi Zhang, Haoran Zheng, Honggeng Ding, Khan Abdul Jabbar, Ling Gao, Guowei Zhao

**Affiliations:** College of Chemistry and Chemical Engineering, Huanggang Normal University, Huanggang 438000, China; legend10086@163.com (Y.Z.);

**Keywords:** solid polymer electrolyte, 3D printing, direct ink writing, ceria quantum dots, Na_2/3_Ni_1/3_Mn_2/3_O_2_ cathode, solid-state battery

## Abstract

Solid polymer electrolytes have been considered as promising candidates for solid-state batteries (SSBs), owing to their excellent interfacial compatibility and high mechanical toughness; however, they suffer from intrinsic low ionic conductivity (lower than 10^−6^ S/cm) and large thickness (usually surpassed over 100 μm or even 500 μm), which has a negative influence on the interface resistance and ionic migration. In this work, ceria quantum dot (CQD)-modified composite polymer electrolyte (CPE) membranes with a thickness of 20 μm were successfully manufactured via 3D printing technology. The CQD fillers can reduce the crystallinity of the polymer, and the oxygen vacancies on CQDs can facilitate the dissociation of ion pairs in the NaTFSI salt to release more free Na^+^, improving the ionic conductivity. Meanwhile, tailoring the thickness of the CPE-CQDs membrane via 3D printing can further promote the migration and transport of Na^+^. Furthermore, the printed NNM//CPE-CQDs//Na SSB exhibited outstanding rate capability and cycling stability. The combination of CQD modification and thickness tailoring through 3D printing paves a new avenue for achieving high performance solid electrolyte membranes for practical application in Na SSBs.

## 1. Introduction

Rechargeable sodium-ion batteries (SIBs) have been considered as promising alternatives to lithium-ion batteries (LIBs) because of their high natural abundance as well as the relatively low cost of Na element, comparable internal components, and similar charge/discharge principles to LIBs. These advantages favor the application of SIBs in large-scale energy storage systems, which is essential for the achievement of carbon neutrality and sustainable development of human society [1]. Recently, a growing number of investigations have focused on exploring and developing high-performance sodium solid-state electrolytes (SSEs) to supersede the currently used organic liquid electrolytes because SSEs can avoid the safety and environmental issues caused by the effumability, easy leakage, and flammability of liquid electrolytes. Moreover, sodium SSEs can serve as functional separators with high Na ionic conductivity between the cathode and anode directly, which is expected to promote the energy density of the energy storage system [2,3,4,5]. The SSEs can be broadly separated into two categories: inorganic solid electrolytes (ISEs) and solid polymer electrolytes (SPEs) [6]. Although Na ISEs, such as Na-β”-Al_2_O_3_, Na_3_Zr_2_Si_2_PO_12_, *t*-Na_3_PS_4_, Na_5.5_PS_4.5_Cl_1.5_, or Na_3_SbS_4_, possess high Na-ion conductivity [7,8], unsatisfactory mechanical stability (brittle and fragile), lack of flexibility, poor interfacial compatibility, and limited processability are the main obstacles hindering the commercialization of ISEs [9,10,11]. Unlike ISEs, SPEs do not have severe interfacial problems and offer high mechanical toughness. Furthermore, the elastic feature of polymer favors the processibility of SPEs, especially for the fabrication of flexible batteries [12]. The polymer electrolyte has been reported by Wright and Armand et al. since the 1970s [13], and numerous polymer-based lithium/sodium solid-state batteries (SSBs) have been designed over the past few decades [14,15,16,17].

Various polymers, such as polyvinylalcohols, polyacrylonitriles, and polyethylene oxide (PEO), have been specifically investigated in Na SSBs. Among them, PEO is an optimal viscoelastic substrate for SPE manufacturing owing to its advantages of low cost, large-scale manufacturing process compatibility, and high mechanical toughness [6,18]. Despite the considerable success of PEO-based SPEs, they suffer from two critical restrictions: (i) the intrinsic low ionic conductivity (lower than 10^−6^ S/cm), which can not meet the requirements for practical applications; and (ii) the thickness of SPE membranes usually surpasses over 100 μm, even 500 μm, which has a negative influence on the interface resistance, ionic migration, and volumetric energy density. Therefore, improving the ionic conductivity and reducing the thickness of the SPEs are of great significance for the practical application.

Incorporating inorganic fillers into SPEs is an effective strategy to overcome the intrinsic poor ionic conductivity of the SPEs since the inorganic fillers can reduce the crystallinity and increase the number of polymer chain segments, promoting the ionic mobility in SPEs [19,20,21]. Moreover, inorganic fillers with oxygen vacancies can accelerate the dissociation of the ion pairs in electrolyte salts (such as LiTFSI or LiClO_4_) and release more free Li^+^ because the oxygen vacancies are capable of serving as the Lewis acid sites to construct strong interaction with the electrolyte salt anion (such as ClO_4_^−^ or TFSI^−^), further enhancing the ionic conductivity of SPEs [22,23,24,25]. Taking the oxygen vacancies into consideration, ceria come into sight as a promising option for the modification of SPEs due to the existence of numerous oxygen vacancies on the surface of CeO_2−*x*_ [26,27,28,29,30,31,32].

Three-dimensional printing, also known as additive manufacturing, is a versatile, eco-efficient, and less-polluting technology [33] capable of rapidly fabricating high-performance polymeric materials for various fields, such as apparel [34], dentistry [35], medicine [36], automotive [37], and soft robots [38]. According to the principles and features, 3D printing technologies can be divided into several typical types, such as direct ink writing (DIW), fused deposition modeling (FDM), selective laser sintering (SLS), stereolithography (SLA), and inkjet printing (IJP) [39]. Among these, DIW is the most promising printing approach owing to its advantages of low cost, convenient operation, and printability for many materials (various materials can be printed as long as they possess suitable rheological properties) [39,40]. In DIW printing, a viscoelastic ink with suitable rheological properties is fed into a nozzle, and then, continuous filaments can be extruded through compressed air and stacked on a building platform. Finally, the extruded deposits are solidified via curing processes such as photopolymerization or thermocuring. In recent years, this technique has advanced rapidly in the manufacture of SPEs for SSBs due to the following advantages [22,41]: (i) the printed thickness of the SPE membrane could be precisely controlled by setting the parameter of DIW operation [11,42,43]; (ii) compared with conventional SSB manufacturing (poor interfacial compatibilities caused by simply pressing SPEs onto the electrodes) [9,14,15], DIW technology can construct an integrity connection between the SPE/electrode interface because both the SPE printing and curing processes (including solvent evaporation and solidification steps) can be achieved directly on the electrode surface, which can effectually lower the interfacial resistance of the SSB [44]. 

In this work, composite polymer electrolyte (CPE) membranes with high ionic conductivity and 20 μm thickness were designed. First, CeO_2−*x*_ quantum dots (CQDs) were prepared using a colloidal synthesis method. Subsequently, CQD-modified polymer electrolyte membranes were fabricated using DIW printing technology. The CQD filler can reduce the crystallinity of the PEO polymer, and the oxygen vacancies on the CQD filler serve as Lewis acid sites to form strong interactions with the TFSI^−^ anion, facilitating the ionic mobility in the CPE. Finally, a thin composite polymer electrolyte was directly printed on the surface of the Na_2/3_Ni_1/3_Mn_2/3_O_2_ cathode using DIW technology for the fabrication of a battery with high electrochemical performance. The combination of incorporating CQD fillers and DIW printing paves the way for the manufacture of thin CPE-CQDs membranes with high electrochemical performance.

## 2. Materials and Methods

### 2.1. Material Synthesis

Ceria quantum dots were synthesized by colloidal synthesis method. A total of 1.5 mmol of cerium (IV) ammonium nitrate (NH_4_)_2_Ce(NO_3_)_6_ and 1.0 g 3-bromopropyl trimethylammonium bromide were dissolved in a mixed solvent of 6 mL oleylamine, 4 mL octadecene, and 2 mL oleic acid under vigorous stirring at 120 °C for 1 h to remove the water and air. Then, the transparent yellow solution was heated to 220 °C for 30 min under a nitrogen atmosphere. After the reaction mixture was cooled to room temperature, the yellow product was readily separated from the bulk solution by centrifugation at 9000 rpm for 10 min. The as-synthesized nanosheets were washed several times with cyclohexane and ethanol, followed by drying in vacuum at 60 °C overnight for further characterization.

For the synthesis of pure PEO membrane, PEO (*M*w = 6 × 10^5^, Aldrich, St. Louis, MO, USA) was added to the anhydrous acetonitrile and mechanically stirred for 24 h to form a homogeneous solution. The solution was cast with a doctor blade on a polytetrafluoroethylene (PTFE) plate and dried in a high-vacuum oven at 60 °C for at least 48 h. Finally, the electrolyte membrane was stored in an argon-filled glove box.

For the synthesis of SPE, NaTFSI (99.5%, Aldrich, St. Louis, MO, USA) and PEO (*M*w = 6 × 10^5^, Aldrich, St. Louis, MO, USA) were added to the anhydrous acetonitrile and mechanically stirred for 24 h to form a homogeneous solution. The molar ratios of Na^+^/EO were 1:20, 1:10, and 1:5. The solution was cast with a doctor blade on a polytetrafluoroethylene (PTFE) plate and dried in a high-vacuum oven at 60 °C for at least 48 h. Finally, the electrolyte membrane was stored in an argon-filled glove box.

For the synthesis of CPE-CQDs, ceria quantum dots, NaTFSI (99.5%, Aldrich, St. Louis, MO, USA), and PEO (*M*w = 6 × 10^5^, Aldrich, St. Louis, MO, USA) were added to the anhydrous acetonitrile and mechanically stirred for 24 h to form a homogeneous solution. The molar ratio of Na^+^/EO was 1:10. The weight ratio of ceria quantum dots/PEO was 1:20. The solution was cast with a doctor blade on a polytetrafluoroethylene (PTFE) plate and dried in a high-vacuum oven at 60 °C for at least 48 h. Finally, the electrolyte membrane was stored in an argon-filled glove box.

For the preparation of CPE-CQDs printing ink, ceria quantum dots, NaTFSI (99.5%, Aldrich, St. Louis, MO, USA) and PEO (*M*w = 6 × 10^5^, Aldrich, St. Louis, MO, USA) were added to the anhydrous acetonitrile and mechanically stirred for 24 h to form a homogeneous solution. The molar ratio of Na^+^/EO was 1:10. The weight ratio of ceria quantum dots/PEO was 1:20. 

For the preparation of CPE-C50 printing ink, 50 nm ceria (99%, Energy), NaTFSI (99.5%, Aldrich, St. Louis, MO, USA), and PEO (*M*w = 6 × 10^5^, Aldrich, St. Louis, MO, USA) were added to the anhydrous acetonitrile and mechanically stirred for 24 h to form a homogeneous solution. The molar ratio of Na^+^/EO was 1:10. The weight ratio of 50 nm ceria/PEO was 1:20. 

For the preparation of CPE-C500 printing ink, 500 nm ceria (99%, Energy), NaTFSI (99.5%, Aldrich, St. Louis, MO, USA), and PEO (*M*w = 6 × 10^5^, Aldrich, St. Louis, MO, USA) were added to the anhydrous acetonitrile and mechanically stirred for 24 h to form a homogeneous solution. The molar ratio of Na^+^/EO was 1:10. The weight ratio of 500 nm ceria/PEO was 1:20. 

The 3D printing process was conducted by 3D printer (S200BH, Xi’an TOP.E, Xi’an, China). The inks were extruded from a nozzle (∅200 μm) by pressurized Ar gas supplied from a pneumatic precision dispenser (Musashi ML-808GX, Japan), where the pressures were set to 200 kPa, the scanning speed of the 3D printing system could be as fast as 120 mm min^−1^, and the gap between nozzle tip and substrate was fixed at around 200 µm. The dispenser was synchronized with an off-the-shelf *xyz*-stage, controlled by Open Source G-code parser and CNC control software (grbl 1.1), while the dispensing conditions were adjusted using custom written Python scripts. The *xyz*-stage was placed in an Ar-filled glove box, and the extrusion of the electrolyte inks was performed in the same Ar-filled glove box. 

For preparation of cathode electrodes, a homogeneous slurry was first obtained by mixing 70 wt% active material, 20 wt% Ketjen black (EC600JD, Japan), and 10 wt% poly(vinyl difluoride) (PVDF, KUREHA, Japan) in N-methylpyrrolidinone (NMP, China, Energy), and then the slurry was printed uniformly onto aluminum foil current collector and dried at 100 °C for 6 h.

For the solid-state battery assembly, CPE-CQDs were directly printed on cathode. The loading mass of the active material was 1–1.2 mg cm^−2^. Then, the volatilization of solvent was conducted by the thermocuring process. Finally, a Na slice was pressed on the opposite side of the printed CPE-CQDs membrane.

For the synthesis of Na_2/3_Ni_1/3_Mn_2/3_O_2_, stoichiometric NaAc, Fe(NO_3_)_3_, NiAc_2_, and MnAc_2_ (Ac = CH_3_COO^−^) were dissolved in 40 mL deionized water to form a dark red solution. A total of 5% excess NaAc was added to compensate the sodium loss at high temperature. The solution was totally evaporated at 80 °C to achieve powder precursor. The precursor was first sintered in O_2_ atmosphere at 350 °C for 4 h and then at 900 °C for another 12 h with a heating rate of 2 °C min^−1^. The obtained sample was quenched to room temperature and then transferred to drying room.

### 2.2. Methods

SEM images were obtained on JEOL JSM-6390 microscope (Tokyo, Japan). TEM images were acquired by a Hitachi HT-7700 transmission electron microscope (Tokyo, Japan) operating at 100 kV. High-resolution TEM (HRTEM) micrographs were obtained with a Philips Tecnai F20 FEG-TEM (CA, USA) operated at 200 kV. Samples for TEM analysis were prepared by drying a drop of cyclohexane solution containing the materials on the surface of a carbon-coated copper grid. X-ray photoelectron spectra were conducted using a PHI 3 Quantera SXM (Tokyo, Japan) instrument equipped with an Al X-ray excitation source (1486.6 eV). Binding energies (BEs) were referenced to the C1s of carbon contaminants at 284.8 eV. For XPS measurements, coin cells were prepared and charged/discharged to the target potential; then, these cells were disassembled in argon-filled glove box to avoid the contact with air. All electrodes were taken out from coin cells and washed with dimethyl carbonate (DMC) three times. After standing for 1 h in vacuum, all electrodes were transferred to the instrument directly for subsequent test. Powder XRD measurements were conducted using a diffractometer equipped with CuKα_1_ radiation (XRD-6010, Shimadzu, Tokyo, Japan) to ascertain the phase compositions of the synthesized powders. Prior to XRD measurements, samples were prepared in a glove box and sealed with polyimide film (Dongguan Meixin Co., Ltd., Guangzhou, China) to shield them from moisture during the measurements. The diffraction data were collected in the 2θ range of 10° to 70° with step widths of 0.01°. The Raman spectra test was performed on an LabRAM Horiba (Tokyo, Japan) single-stage spectrometer with a CCD Symphony detector (Jobin Yvon, Paris, France) with 2048 horizontal pixels. Laser power (the 514 nm line of an Ar^+^ laser, Tokyo, Japan) on the sample was typically less than 0.1 mW. Spectral resolution was 3.0 cm^−1^. Single spectra were collected from the regions with a cross section of approximately 1 μm.

The ionic conductivities of the electrolyte membrane were measured using the AC impedance technique with a stainless steel electrode symmetrical cell (stainless steel/SE/stainless steel), employing pressurizable sealing molds and measured at 25 °C. This measurement was repeated two or three times, applying 15 mV in the frequency range of 7 MHz to 1 Hz, utilizing a potentiostat electrochemical interface (Bio-Logic SAS, VSP-300, Paris, France). The CQD-modified composite polymer electrolyte (CPE) membrane with a diameter of 10 mm was tested under a pressure of 4 MPa using a polyaryletherketone mold.

A solid-state battery employing the obtained CPE-CQDs membrane with the highest ionic conduction as the electrolyte was assembled under a pressure of 4 MPa within an Ar-filled glove box to evaluate its charge–discharge performance. The charge–discharge measurements were conducted at 25 °C, between 2.4 and 4.1 V, using the LANHE CT2001A charge–discharge system (Wuhan LAND Electronics Co., Wuhan, China).

## 3. Results

### 3.1. Physical Characterization of CQDs

The CQDs were synthesized by the colloidal synthesis method, and the detailed experimental section is described in the Materials and Methods section. All the peaks in the XRD patterns of CeO_2−*x*_ could be indexed to JCPDS: 43-1002 (Figure 1a), confirming the cubic fluorite-type structure with the Fm-3m space group. Detailed Rietveld refinement data were calculated and are presented in Appendix A. The size of the monodispersed CQDs was approximately 5 nm (Figure 1b,c). The (200) plane with an interplanar distance of 0.26 nm could be clearly observed in the high-resolution transmission electron microscopy (HRTEM) image (Figure 1d). The corresponding selected area electron diffraction (SAED) patterns of the CQDs displayed the (111), (200), (220), and (311) diffraction rings (Figure 1e). The Ce 3d core spectra are displayed in Figure 1f, and the peaks at 903 and 884 eV could be assigned to Ce^3+^ 3d states. The presence of Ce^3+^ valence states was attributed to the existence of oxygen vacancies in CeO_2−*x*_, and the concentration of Ce^3+^ in CeO_2−*x*_ was 19%, which could be obtained by calculating the ratio of the Ce^3+^ integrated area to the total area of the Ce 3d core spectra [32].

### 3.2. Physical and Electrochemical Characterizations of the Electrolyte Membrane

To investigate the effect of the Na^+^/EO molar ratio on the SPE membranes, Raman spectroscopy and Electrochemical Impedance Spectroscopy (EIS) tests were carried out. With an increase in the Na^+^/EO ratio from 0 to 1/10, the intensity of the specific peak around 860 cm^−1^ decreased gradually, indicating the complexing of Na^+^ and ether oxygen (Appendix A). The formation of Na polymer solvates can disorder the crystalline structure of PEO and enhance the ionic conductivity of the SPE [45]. However, no obvious change was observed when the Na^+^/EO ratio was further increased to 1/5. In addition, the Na^+^ conductivity of the SPE membranes with a thickness of 100 μm was measured using EIS. The Na^+^/EO = 1/10 membrane displayed a smaller semicircle in the impedance spectra than that of the Na^+^/EO = 1/20 membrane, while the semicircle showed no obvious change when the Na^+^/EO ratio increased to 1/5 (Appendix A). Therefore, the optimal ionic conductivity of the SPE membrane was determined as Na^+^/EO = 1/10.

To further modify the electrochemical properties of the SPE membrane, the as-synthesized CQDs were employed as inorganic fillers for the preparation of CPE membrane. The SPE membrane with a Na^+^/EO ratio of 1/10 was slightly curved and transparent (Figure 2a). In contrast, the membrane became semitransparent and flat after the addition of 5% CQDs (Figure 2b). The crystallinity of PEO can affect the ionic conductivity of electrolytes [46], and the crystallinity of PEO in pure PEO, SPE, and CPE-CQDs membranes was investigated using XRD (Figure 2c). The characteristic peaks of the pure PEO membrane in the XRD pattern were strong and sharp, whereas the intensity of these characteristic peaks became weaker after the introduction of NaTFSI (Na^+^/EO = 1/10), which implies a lower crystallinity of PEO in the SPE membrane. Moreover, the same tendency was clearly observed in the XRD pattern of the CPE-CQDs membrane, demonstrating that incorporating CQD fillers into the SPE could further reduce the crystallinity of PEO. 

Subsequently, the pure PEO, SPE, and CPE-CQDs membranes were investigated using Raman spectroscopy (Figure 2d). No peak was detected in the pure PEO membrane, whereas the peak correlating with free TFSI^-^ species was observed at 738 cm^−1^ in the SPE membrane. After the introduction of CQDs into SPE, a new peak, recognized as TFSI^−^-CQD coordination, appeared at 751 cm^−1^ [47]. Such interaction between TFSI^−^ anions and the oxygen vacancy on CQDs could further facilitate the dissociation of the ion pairs in NaTFSI salt and release more free Na^+^, enhancing the mobility of Na^+^ in the membrane [22,23,24,25]. 

The Na^+^ conductivities of the SPE and CPE-CQDs membranes with a thickness of 100 μm were measured by EIS. The impedance spectra of the CPE-CQDs membrane exhibited charge–transfer resistance about 300 Ω, which was smaller than that of the SPE membrane (approximately, 600 Ω). The calculated Na^+^ conductivities of SPE and CPE-CQDs membranes were 2.1 × 10^−4^ S/cm and 4.2 × 10^−4^ S/cm, indicating that incorporating CQD fillers into SPE could improve the Na^+^ conductivity in the membrane. LSV was used to test the electrochemical stability windows of the SPE and CPE-CQDs membranes (Appendix A). The LSV curve of the SPE membrane showed obvious oxidation from 4.2 V, while the CPE-CQDs membrane remained stable at over 4.5 V, indicating that the strong interaction between the oxygen vacancy of CQDs and TFSI anion could prevent the decomposition of sodium salt anion, expanding the electrochemical stability window [48,49,50]. Galvanostatic cycling tests were conducted in Na foil symmetric cell configurations to investigate the interfacial stability between the electrolyte membrane and Na electrodes. The symmetric cell built with the SPE and CPE-CQDs membranes was charged and discharged for 1 h, respectively, under a current density of 0.05 mA/cm^2^. The SPE membrane displayed a hard short within the first hour and a high overpotential of 0.04 V for Na^+^ plating/stripping after 30 h. In contrast, the overpotential of the CPE-CQDs membranes (0.02 V) remained almost the same over 120 h, indicating the excellent interfacial stability between the CPE-CQDs membrane and Na electrodes. Therefore, incorporating CQD fillers into SPE can efficiently improve the electrochemical performance of the electrolyte membrane.

### 3.3. Physical and Electrochemical Characterizations of the 3D-Printed Electrolyte Membrane

DIW was employed to print the electrolyte membrane. The printing process was described in the Materials and Methods Section. Figure 3a shows the structure of the 3D printer used for the DIW. The precursor ink was fed into the nozzle, and then continuous filaments were extruded through compressed air and stacked on the teflon building platform (Figure 3b). The extruded deposits solidified directly via thermocuring (Figure 3c). To verify the influence of CeO_2−*x*_ size on the DIW-printed electrolyte membranes, 5% CQDs, CeO_2−*x*_-50 nm, and CeO_2−*x*_-500 nm were added as fillers into the precursor ink for the printing of CPE membranes (denoted as CPE-CQDs, CPE-C50, and CPE-C500). The XRD patterns of the CQDs, C50, and C500 samples are shown in Appendix A. All the samples exhibited a pure CeO_2−*x*_ phase. The peaks of C50 and C500 samples were sharp and narrow, while the peaks of CQD sample broadened due to the particle size effects [51]. Figure 3d shows the smooth SPE membrane printed by DIW technology with no inorganic filler. After the addition of 5% CQDs, the printed CPE-CQDs membrane still remained smooth and intact (Figure 3e). Nevertheless, some slit pores appeared on the printed membrane after the addition of 5% C50 (Figure 3f). When 5% C500 was added, the slit pores in the printed membrane increased (Figure 3g). SEM images showed that the CQDs were well-dispersed in the SPE membrane (Figure 3h,i), whereas the C50 and C500 fillers were close to each other with severe aggregation (Figure 3j,k). Therefore, the particle size of CeO_2−*x*_ filler could affect the integrity of the printed membrane, which could be attributed to the change in the ink’s rheological and surface tension by the aggregation of CeO_2−*x*_ fillers [52]. 

In addition, CPE-CQDs membranes with different thicknesses (54 μm, 105 μm, and 204 μm) were printed via DIW (Appendix A). The Na^+^ conductivities of these printed membranes were measured using EIS (Appendix A). The CPE-CQDs membrane with a thickness of 54 μm exhibited the lowest charge–transfer resistance of 150 Ω than that of 105 and 204 μm, indicating that reducing the thickness of the electrolyte membrane could promote the migration and transport of Na^+^. It should be noted that the printed membrane with a thickness lower than 50 μm was difficult to strip from the teflon platform, whereas thinner CPE-CQDs membranes could be printed directly on the surface of the electrode to achieve higher electrochemical properties.

### 3.4. Electrochemical Performance of the 3D-Printed NNM//CPE-CQDs//Na SSB

To verify the practical application of DIW technology for the preparation of SSBs with thin CPE-CQDs membranes, Na_2/3_Ni_1/3_Mn_2/3_O_2_ (NNM) was employed as the cathode material because of its outstanding electrochemical stability [53]. The XRD pattern in Figure 4a could be indexed to a hexagonal structure with a space group of *P6_3_/mmc* (JCPDS: 54-0894). No other impurity phases were detected, indicating that the NNM cathode was phase pure. The SEM image (Appendix A) exhibited the layered structure of the NNM sample. The HRTEM image in Figure 4b represents a Ni/Mn layer in the NNM layer. The 100 plane of the NNM layer can be clearly observed with an interplanar distance of 0.25 nm. The corresponding FFT (inset of Figure 4b) confirms the hexagonal phase. First, the NNM cathode was printed via DIW on an Al current collector followed by drying. Then, the CPE-CQDs precursor ink was printed via DIW on the surface of the NNM cathode (Figure 4c), and a thin CPE-CQDs membrane with a thickness of only 20 μm was formed directly after the thermocuring processes (Figure 4d). Finally, a Na metal slice was pressed on the opposite side of the printed CPE-CQDs membrane for the assembly of the NNM//CPE-CQDs//Na SSB (approximately 4 cm × 4 cm). The NNM//CPE-CQDs//Na SSB exhibited an open-circuit voltage of 3.2 V and could light up an LED bead (Figure 4e,f).

The electrochemical properties of the NNM//CPE-CQDs//Na SSB are shown in Figure 5. The cyclic voltammetry (CV) curves of the NNM//CPE-CQDs//Na SSB presented redox peaks at ∼3.3 and 3.6 V, indicating the existence of two different Na^+^ vacancy ordering arrangements in NNM cathode (Figure 5a) [54]. The charge/discharge profiles of the NNM//CPE-CQDs//Na SSB displayed two plateaus at ∼3.3 and 3.6 V with a discharge capacity about 80 mAh/g (Figure 5b). The rate capability of the NNM//CPE-CQDs//Na SSB was tested at different current densities. As shown in Figure 5c, the NNM//CPE-CQDs//Na SSB delivered a discharge capacity of 80 mAh g^−1^ at 0.05 C, and 54% (approximately 43 mAh g^−1^) of the initial discharge capacity was maintained at 2 C. When back to 0.05 C, the discharge capacity recovered to 79 mAh/g. The cycling performance of NNM//CPE-CQDs//Na SSB was evaluated at 0.2 C (Figure 5d). A discharge capacity of approximately 80 mAh/g was achieved during the initial cycle, and 92% (74 mAh/g) of its initial discharge capacity could be maintained after 200 cycles. The comparison results with other related works are listed in Appendix A. The outstanding rate capability and the cycling stability revealed that the DIW-printed thin CPE-CQDs membrane (20 μm) possessed the ability for practical application in sodium SSBs.

## 4. Discussion

In summary, a thin CPE-CQDs membrane with a thickness of 20 μm was successfully manufactured using 3D printing technology. Incorporating CQDs into the SPE membrane could reduce the crystallinity and increase the number of polymer chain segments, promoting Na^+^ mobility in the SPE membrane. In addition, DIW printing was used to tailor the thickness of the CPE-CQDs membranes, further promoting the migration and transport capability of Na^+^. The rationally designed thin CPE-CQDs membrane could be directly utilized for the assembling of NNM//CPE-CQDs//Na configuration, which exhibited an outstanding rate capability and cycling stability. The combination of CQDs modification and thickness tailoring through DIW opens a new avenue for manufacturing high-performance solid electrolyte membranes for the practical application of SSBs.

## Figures and Tables

**Figure 1 polymers-16-01707-f001:**
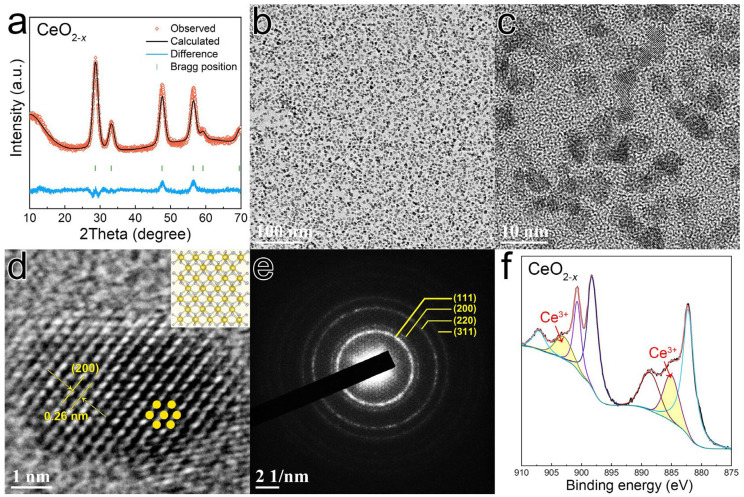
(**a**) XRD patterns and Rietveld refinement of the CQDs. (**b**,**c**) TEM images of the monodispersed CQDs. (**d**) HRTEM image of CQDs. (**e**) Corresponding SAED patterns of CeO_2−*x*_. (**f**) XPS spectra of the Ce 3d core in the CQDs.

**Figure 2 polymers-16-01707-f002:**
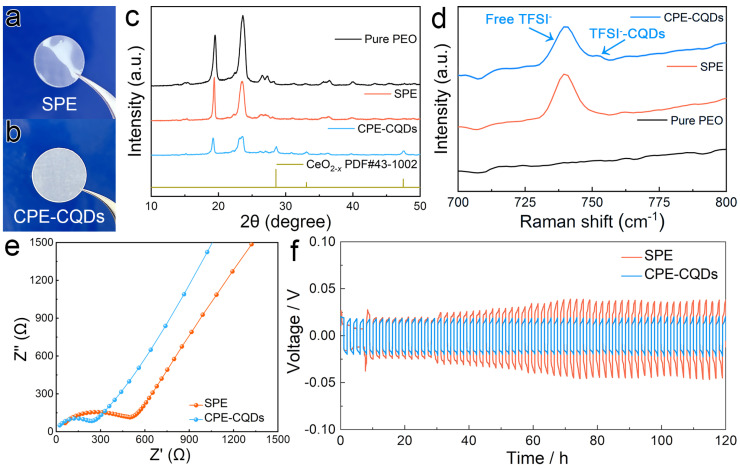
(**a**,**b**) The digital pictures of the SPE and CPE-CQDs membranes. (**c**) XRD patterns of pure PEO, SPE, and CPE-CQDs membranes. (**d**) Raman spectra of pure PEO, SPE, and CPE-CQDs membranes. (**e**) Impedance plots of the SPE and CPE-CQDs membranes. (**f**) Long-term galvanostatic cycling curves of Na symmetric cells built with SPE and CPE-CQDs at a current density of 0.05 mA/cm^2^.

**Figure 3 polymers-16-01707-f003:**
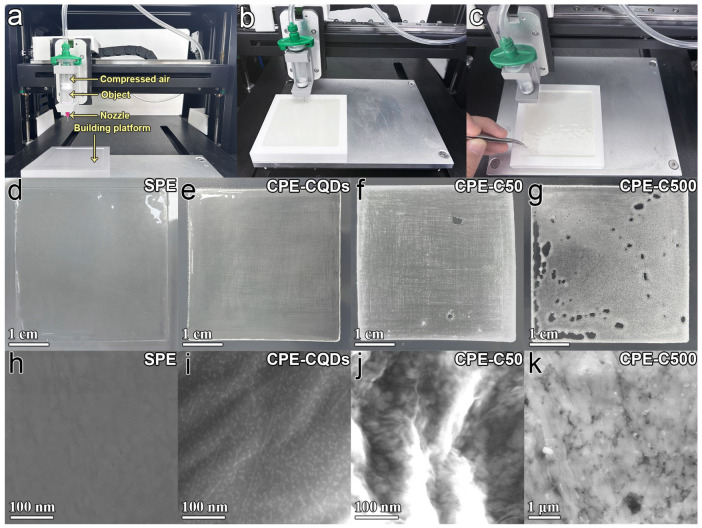
(**a**) The constructions of 3D printer for DIW. (**b**,**c**) The digital picture of the printed membrane before and after thermocuring. (**d**–**g**) The digital images of SPE, CPE-CQDs, CPE-C50, and CPE-C500 membranes. (**h**–**k**) The SEM images of the SPE, CPE-CQDs, CPE-C50, and CPE-C500 membranes.

**Figure 4 polymers-16-01707-f004:**
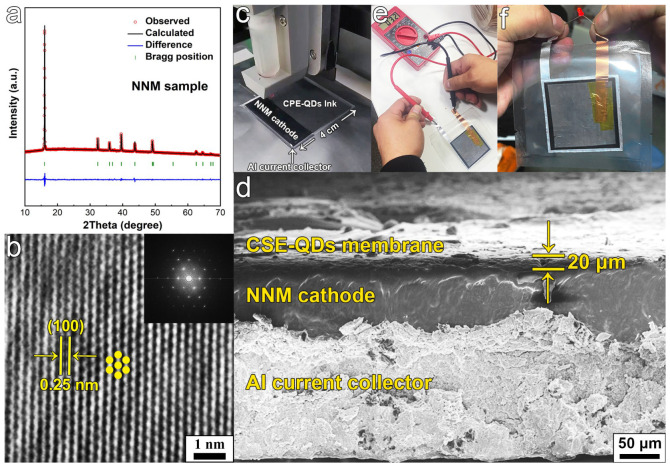
(**a**) XRD patterns of the NNM sample. (**b**) HRTEM image and FFT patterns (inset) of the NNM sample. (**c**) Digital picture of NNM cathode and CPE-CQDs membrane using DIW printing technology. (**d**) SEM image of the printed cathode and electrolyte membrane. (**e**,**f**) Digital picture of NNM//CPE-CQDs//Na SSB working at room temperature to illuminate an LED bead.

**Figure 5 polymers-16-01707-f005:**
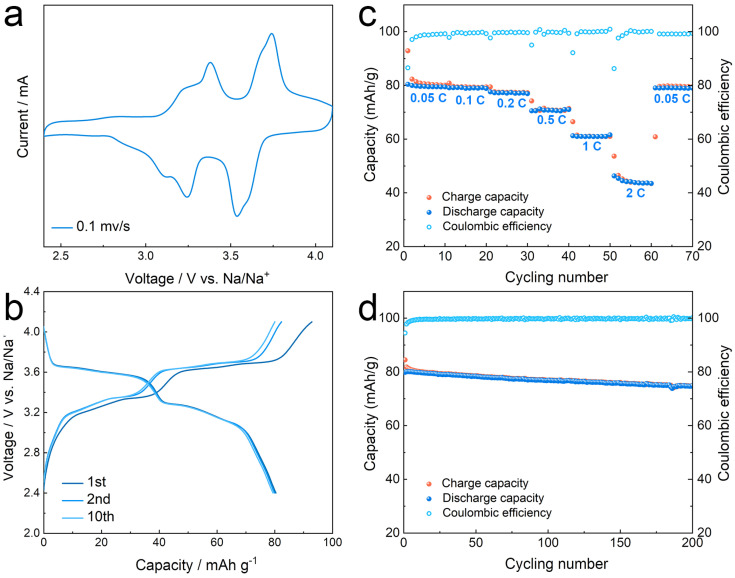
(**a**) Cyclic voltammetry of the NNM//CPE-CQDs//Na SSB. (**b**) Galvanostatic charge/discharge curves of NNM//CPE-CQDs//Na SSB at 0.05 C. (**c**) Rate capability of NNM//CPE-CQDs//Na SSB. (**d**) Cycling stability of NNM//CPE-CQDs//Na SSB.

## Data Availability

Data are contained within the article.

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
