# Peer review of "Ceria Quantum Dot Filler-Modified Polymer Electrolytes for Three-Dimensional-Printed Sodium Solid-State Batteries"

_polymers, 2024, doi:10.3390/polym16121707_

Round 1

Reviewer 1 Report

Comments and Suggestions for Authors

This is very few of the MDPI Polymers works that I feel pretty well-written and organized with very meaningful results and detailed characterization data. I cannot imagine if such kind of high quality work cannot be published in MDPI Polymers. However, all of my comments have to be addressed accordingly before it becomes a final publication in MDPI Polymers.

1. The authors stated that adding O vacant fillers can promote Li+ dissociation. What did the authors thinking about: reduced crystallinity of PEO vs. Li-salt dissociation to give an enhanced conductivity and performance? Which factor can lead to more obvious impact?

2. How high the temperature did the authors use for running the full cell performance evaluation? Since no temperature information is given, I am assuming it is at room temperature. If not, did the authors see any temperature effect which accelerate the side reactions happening at the electrode-electrolyte interface? How would the authors make 20 um thick separator? Through Spin coating?

3. Given the emphasis on the PEO-based SPE that the authors gave in the introduction part, both or one of the following significant work regarding the PEO-based electrolyte should also be considered highlighting in the Page 2, lines 46-58: Energy Storage Materials, 57, 429-459, 2023 (this work highlighted the significant research progress in PEO-based electrolytes), and for example: Scientific Reports 10, Article number: 4390 (2020), which showed high voltage designs that may be relevant to the authors' work. 

4.  Can the authors include a table to compare your results with the state-of-the art published results?  This should be in terms of capacity retention, C-rate, mass loading, testing temperatures, cathode electrodes employed.

5. Why did the authors use 3D printing instead of conventional solid-state methods to prepare the fillers, which seems to me the latter is more scalable and should give a comparable performance? Explanation should be provided in the introduction section. 

Author Response

Our Reply to the Reviewers’ Comments and Questions

Reviewer #1:

Comments and questions:

This is very few of the MDPI Polymers works that I feel pretty well-written and organized with very meaningful results and detailed characterization data. I cannot imagine if such kind of high quality work cannot be published in MDPI Polymers. However, all of my comments have to be addressed accordingly before it becomes a final publication in MDPI Polymers:

(1) The authors stated that adding O vacant fillers can promote Li+ dissociation. What did the authors thinking about: reduced crystallinity of PEO vs. Li-salt dissociation to give an enhanced conductivity and performance? Which factor can lead to more obvious impact?

(2) How high the temperature did the authors use for running the full cell performance evaluation? Since no temperature information is given, I am assuming it is at room temperature. If not, did the authors see any temperature effect which accelerate the side reactions happening at the electrode-electrolyte interface? How would the authors make 20 um thick separator? Through Spin coating?

(3) Given the emphasis on the PEO-based SPE that the authors gave in the introduction part, both or one of the following significant work regarding the PEO-based electrolyte should also be considered highlighting in the Page 2, lines 46-58: Energy Storage Materials, 57, 429-459, 2023 (this work highlighted the significant research progress in PEO-based electrolytes), and for example: Scientific Reports 10, Article number: 4390 (2020), which showed high voltage designs that may be relevant to the authors' work.

(4) Can the authors include a table to compare your results with the state-of-the art published results? This should be in terms of capacity retention, C-rate, mass loading, testing temperatures, cathode electrodes employed.

(5) Why did the authors use 3D printing instead of conventional solid-state methods to prepare the fillers, which seems to me the latter is more scalable and should give a comparable performance? Explanation should be provided in the introduction section. 

The responses to the questions:

Question 1:

The authors stated that adding O vacant fillers can promote Li+ dissociation. What did the authors thinking about: reduced crystallinity of PEO vs. Li-salt dissociation to give an enhanced conductivity and performance? Which factor can lead to more obvious impact?

Response: We appreciate the reviewer for this kind question. Decreasing the crystallinity and increasing Li-salt dissociation by oxygen vacancies, both two modifications can improve the ion conductivity in the polymer, whereas their mechanisms are totally different:

On one hand, crystallinity is a fundamental characteristic of polymeric systems. In the case of polyethylene oxide (PEO)—the archetypal solid polymer electrolyte (SPE) for Li+ conduction—the presence of polymer crystallites which appear below the melting point usually leads to a low ionic conductivity since Li+ salts in SPE would be excluded from polymer crystallites which are constituted of tightly packed polymer chains. The mobility of Li+ mainly depends on the movement of polymer chain segments at the grain boundary and amorphous phase region, and the ion conductivity through the grain boundary and amorphous phase region is much higher than that through the crystalline lamellae. Thus, control over the polymer crystallinity is therefore of paramount importance to achieve high ionic conductivity.

       On the other hand, oxygen vacancies on the surface of these nanofillers act as Lewis acid sites that interact with the anions of the salts to free Li-ions.

Therefore, it is very hard to say which modification strategy is better based on the perspective of mechanism. However, I think reducing crystallinity leads to more obvious impact based on the electrochemical test results from this reference (doi.org/10.1002/smll.202102039). The author used normal Al2O3 and oxygen vacancy contained Al2O3-δ to modify the polymer electrolyte. The normal Al2O3 can reduce the crystallinity of the PEO and increase the ionic conductivity of the polymer electrolyte from 1×10-6 to 1.36×10-4 S/cm, almost 100 times promotion by reducing crystallinity. In contrast, with the assistance of oxygen vacancy, adding Al2O3-δ increase the ionic conductivity from 1.36×10-4 S/cm to 3.31×10-4 S/cm, about 3 times promotion. Thus, reducing crystallinity leads to more obvious impact.

Question 2:

How high the temperature did the authors use for running the full cell performance evaluation? Since no temperature information is given, I am assuming it is at room temperature. If not, did the authors see any temperature effect which accelerate the side reactions happening at the electrode-electrolyte interface? How would the authors make 20 um thick separator? Through Spin coating?

Response: We appreciate the reviewer for this kind question. First, the temperature is about 30°C, which is very close to the room temperature. Many references use 60~70°C to test the electrochemical performance of the cell (they usually just press the electrolyte membrane onto the cathode, which usually results in poor interfacial resistance) since the high temperature can modify the interfacial contact. However, in this work, electrolytes can be directly printed onto the electrode surfaces without the need for any surface preparation, which can significantly reduce the interfacial resistance. Here is a similar reference using 3D-printing to fabricate cell and testing cell under room temperature (doi.org/10.1002/adma.201800615).

In addition, we make the 20-um thick electrolyte membrane through 3D-printing technology. For the solid-state battery assembly, CPE-CQDs was directly printed on cathode via DIW process. The DIW is one type of 3D-printing technology, which is an effective strategy to control the thickness of polymer membrane.

Question 3:

Given the emphasis on the PEO-based SPE that the authors gave in the introduction part, both or one of the following significant work regarding the PEO-based electrolyte should also be considered highlighting in the Page 2, lines 46-58: Energy Storage Materials, 57, 429-459, 2023 (this work highlighted the significant research progress in PEO-based electrolytes), and for example: Scientific Reports 10, Article number: 4390 (2020), which showed high voltage designs that may be relevant to the authors' work.

Response: We appreciate the reviewer for this kind suggestion. These two references have been cited as reference 16 and reference 17. 

Based on the information above, we added these two references in the manuscript with red colour.

Question 4:

Can the authors include a table to compare your results with the state-of-the art published results? This should be in terms of capacity retention, C-rate, mass loading, testing temperatures, cathode electrodes employed.

Response: We appreciate the reviewer for this kind suggestion. We have searched the related reference, which contained the Na2/3Ni1/3Mn2/3O2 (NNM) cathode and solid-state electrolyte. The comparison results are listed in table 1. We have added this table in supporting information. We also added the sentence “The comparison results with other related works were listed in Table S2.” into the manuscript with red colour.

Table 1. The comparation of sodium solid state battery.

Cathode

Electrolyte

C-Rate

Cycle number

Residual Capacity (mAh/g)

Retention (%)

Mass loading

mg/cm2

Size

Test °C

ref

Na2/3Ni1/3Mn2/3O2

PEO/NaFSI

0.1 C

100

140

99.91

2

Coin cell

20

1

Na2/3Ni1/3Mn2/3O2

P(EO/MEEGE)

1/30 C

30

69

93

2.1

Coin cell

60

2

Na3V2(PO4)3

NaFSI/PEO/Al2O3

0.25 mA/cm2

2000

84

93

3

Coin cell

80

3

Na3V2(PO4)3

NaClO4/PEO/Al2O3

2 C

1000

84

87.5

2.2

Coin cell

80

4

Na3V2(PO4)3

NaClO4/PMA/α-Al2O3

0.5

350

80

94.1

Not mention

Coin cell

70

5

Na3V2(PO4)3

NaClO4/PEO/CQDs

1

200

50

45.1

2

Coin cell

60

6

Na3V2(PO4)3

NaTFSI/PEO/Na3.4Zr1.8Mg0.2Si2PO12

0.1

120

100

99

2.5

Coin cell

80

7

Na2MnFe(CN)6

NaClO4/PEO/Na3Zr2Si2PO12

0.5

300

124

72

3

Coin cell

60

8

Na2/3Ni1/3Mn2/3O2

PEO/NaTFSI/Ceria

0.2 C

200

74

92

1~1.2

4 x 4 cm

30

This work

Question 5:

Why did the authors use 3D printing instead of conventional solid-state methods to prepare the fillers, which seems to me the latter is more scalable and should give a comparable performance? Explanation should be provided in the introduction section. 

Response: We appreciate the reviewer for this kind question. We do not use 3D printing to synthesis the fillers. The 3D-Priting was used to prepare polymer membrane. The ceria quantum dots fillers was prepared via colloidal synthesis method, which was described in the Materials and Methods section. The detail is: 1.5mmol of cerium(IV) ammonium nitrate (NH4)2Ce(NO3)6 and 1.0 g 3-bromopropyl trimethylammonium bromide were dissolved in a mixed solvent of 6 ml oleylamine, 4 ml octadecene and 2 ml oleic acid under vigorous stirring at 120 °C for 1h to remove the water and air. Then this transparent yellow solution was heated to 220 °C for 30 min under a nitrogen atmosphere. After the reaction mixture was cooled to room temperature, the light yellow products could be readily separated from the bulk solution by centrifugation at 9000 rpm for 10 min. The as-synthesized nanosheets were washed several times with cyclohexane and ethanol, followed by drying in vacuum at 60℃ overnight for further characterization.

  1. Roscher, D.; Kim, Y.; Stepien, D.; Zarrabeitia, M.; Passerini, S., Solvent‐free Ternary Polymer Electrolytes with High Ionic Conductivity for Stable Sodium‐based Batteries at Room Temperature. Batteries & Supercaps 2023, 6 (9), e202300092.
  2. Tatara, R.; Suzuki, H.; Hamada, M.; Kubota, K.; Kumakura, S.; Komaba, S., Application of P2-Na2/3Ni1/3Mn2/3O2 Electrode to All-Solid-State 3 V Sodium(-Ion) Polymer Batteries. Journal of Physical Chemistry C 2022, 126 (48), 20226-20234.
  3. Liu, L.; Qi, X.; Yin, S.; Zhang, Q.; Liu, X.; Suo, L.; Li, H.; Chen, L.; Hu, Y.-S., In Situ Formation of a Stable Interface in Solid-State Batteries. ACS Energy Lett. 2019, 4 (7), 1650-1657.
  4. Gao, R.; Tan, R.; Han, L.; Zhao, Y.; Wang, Z.; Yang, L.; Pan, F., Nanofiber networks of Na3V2(PO4)3 as a cathode material for high performance all-solid-state sodium-ion batteries. J. Mater. Chem. A 2017, 5 (11), 5273-5277.
  5. Zhang, X.; Wang, X.; Liu, S.; Tao, Z.; Chen, J., A novel PMA/PEG-based composite polymer electrolyte for all-solid-state sodium ion batteries. Nano Research 2018, 11 (12), 6244-6251.
  6. Ma, C.; Dai, K.; Hou, H.; Ji, X.; Chen, L.; Ivey, D. G.; Wei, W., High Ion-Conducting Solid-State Composite Electrolytes with Carbon Quantum Dot Nanofillers. Adv. Sci. 2018, 5 (5), 1700996.
  7. Zhang, Z.; Zhang, Q.; Ren, C.; Luo, F.; Ma, Q.; Hu, Y.-S.; Zhou, Z.; Li, H.; Huang, X.; Chen, L., A ceramic/polymer composite solid electrolyte for sodium batteries. J. Mater. Chem. A 2016, 4 (41), 15823-15828.
  8. Yu, X.; Xue, L.; Goodenough, J. B.; Manthiram, A., A High-Performance All-Solid-State Sodium Battery with a Poly(ethylene oxide)–Na3Zr2Si2PO12 Composite Electrolyte. ACS Mater. Lett. 2019, 1 (1), 132-138.

Reviewer 2 Report

Comments and Suggestions for Authors

Comments on the manuscript “Ceria Quantum Dots Fillers Modified Polymer Electrolytes for 2 3D-Printed Sodium Solid-State batteries” (polymers 3057789)

      The topic of the manuscript is undoubtedly interesting for readers of “Polymers”. The manuscript describes the manufacturing solid polymer electrolyte with Na+ conductivity. The combination of such approaches as using ceria quantum dots as a filler, and 3D printing resulted in interesting features. At the same time, the manuscript needs in minor revision.

            1. Ceria quantum dots were synthesized from solution of (NH4)2Ce(NO3)6 in a solvent with sophisticated composition. It would be worth explaining why this very synthesis method was chosen. It is not clear whether this method is an original development of the authors or it was borrowed from the literature. It is written (lines 201, 202):  The CQDs were synthesized by the colloidal synthesis method, and the detailed experimental section is described in the Supporting Information. However, in reality the Supporting Information contains no description of the synthesis. 

2. What was a current collector of positive electrode (cathode), and how the active material was applied onto this current collector?      

3. It is desirable to note whether the results of conductivity measurements were dependent on a pressure. Why the pressure 4 MPa was chosen?

4. It is desirable to discuss the possible causes of lowered rate of CPE-CQDs anodic oxidation in comparison with SPE (Figure S3).  

5. The dimensionality must be written at the Y-axis in Figure 5(c).
